



# The influence of tropical Indian Ocean warming and Indian Ocean Dipole on the surface chlorophyll concentration in the eastern Arabian Sea

Syam Sankar[1], Anoop Thondithala Ramachandran[2], Kemgang Ghomsi Franck Eitel[3,4], Dmitry Kondrik[5],
Radharani Sen[6], Ramesh Madipally[2] and  Lasse H. Pettersson[7]

[1]Nansen Environmental Research Centre India (NERCI), Kochi, 68206, India
[2]ESSO-National Centre for Earth Science Studies (ESSO-NCESS), Thiruvananthapuram, 695011, India
[3]Department of Physics, The University of Yaoundé I, P.O. Box 812, Yaoundé, Cameroon
[4]Nansen Scientific Society (NSS), Bergen, N-5006, Norway
[5]Nansen International Environmental and Remote Sensing Center (NIERSC), St. Petersburg, 199034, Russia
[6]Indian Institute of Technology (IIT), Kharagpur, 721302, India
[7]Nansen Environmental and Remote Sensing Center (NERSC), Bergen, Norway

*Correspondence to*: Syam Sankar (syamsankar1@gmail.com)

**Abstract.** This study examines the role of increased Indian Ocean warming and positive Indian Ocean Dipole (IOD) events on the surface chlorophyll concentration in the Eastern Arabian Sea (EAS) during the period 1998 to 2014. Remotely sensed surface chlorophyll concentration - during the month of October when IOD strength is maximum - at three selected areas in the EAS, viz., the central eastern Arabian Sea (CEAS,  73° E-76° E, 13° N - 18° N), south eastern Arabian Sea (SEAS, 74° E - 77° E, 8° N - 13° N) and the southern tip of India (TIP, 74° E - 78° E, 5° N - 8° N) shows a steady decreasing trend, though not statistically significant. The EAS also has a higher warming trend when compared to the western basin during the study period. Our analysis has shown that in the EAS, local surface winds, remote forcing by equatorial winds and the surface and sub-surface oceanic circulation features are less (respectively more) favorable for inducing coastal upwelling during positive (respectively negative) IOD years. The Dipole Mode Index (DMI) and surface chlorophyll concentration in the EAS is significantly and negatively correlated, pointing to the fact that in the event of occurrence of frequent positive IOD years under a global warming regime, the surface chlorophyll concentration is likely to decrease during fall.

## 1 Introduction

The chlorophyll distribution in the tropical Indian Ocean is strongly influenced by the seasonally reversing monsoon winds (Wiggert et al., 2006). During the northeast monsoon (NEM), northeasterly wind-driven convective mixing results in winter blooms over the central and northern Arabian Sea (Madhupratap et al., 1996). During the southwest monsoon (SWM) coastal upwelling, Ekman pumping and horizontal advection result in increased chlorophyll concentrations in the western Arabian Sea off the east African coast (Latasa and Bidigare, 1998). Bauer et al. (1991) showed that cyclonic (positive) wind-stress curl occurring northwest of the Findlater Jet (monsoon low level jet) induces a divergent Ekman transport in the upper-ocean



and drive open ocean upwelling through upward Ekman pumping in the central Arabian Sea. Kumar et al. (2001) reported that high biological productivity seen in the central Arabian Sea during SWM is driven by Ekman pumping and lateral advection of upwelled waters from the coastal region of Arabia. In August and September, a phytoplankton bloom develops in the central Arabian Sea region as the SWM declines. Banse (1987) suggested that an important nutrient source for this bloom is vertical mixing induced by the strong winds, which brings nutrient-rich subsurface water into the euphotic zone. Another nutrient source of this bloom is the water upwelled near the Arabian coast that is subsequently advected offshore into the central Arabian Sea. Detrainment blooms also occur in the central Arabian Sea during fall and throughout most of the basin during spring. These blooms are intense, short-lived events that develop when the mixed layer thins abruptly, thereby quickly increasing the depth-averaged light intensity available for phytoplankton growth (McCreary et al., 1996). Along the south west coast of India, the prominent upwelling region associated with the SWM is found to be between 8° N and 14° N (Smitha et al., 2014). Jayaram et al. (2013) showed that increase in the alongshore component of wind along the south west coast of India resulted in increased Ekman mass transport which, in turn lead to an increase in the surface chlorophyll concentration. Smitha et al. (2008) are of the view that upwelling processes off the southern tip and west coast of India are highly localized features with different forcing mechanisms and they cannot be treated as a uniform wind-driven upwelling system. A recent study by Amol (2018) has highlighted the influence of Rossby waves on the westward propagation of the surface chlorophyll in the southeastern Arabian Sea. Any variability in chlorophyll concentration in the southeastern Arabian Sea can also influence the landings of the Indian oil sardine, a major pelagic fish landed in the area (Menon et al., 2019).

Two important physical modes of variability in the inter-annual scale exist in the Indian Ocean that have a pronounced influence on the variability of the marine biological productivity in the basin, viz., the indigenous Indian Ocean Dipole (IOD) (Saji et al., 1999; Webster et al., 1999) and the well-known El Niño–Southern Oscillation (ENSO) (Yu et al., 1999; Chambers et al., 1999). The IOD is a coupled ocean–atmosphere phenomenon originating in the Indian Ocean characterized by basin-scale sea surface temperatures (SSTs) and wind anomalies. A positive (negative) phase of the IOD results in anomalous (warming) cooling of SST in the eastern tropical Indian Ocean (off Java Island) whereas the western tropical Indian Ocean (off the east coast of Africa) tends to experience an anomalous SST warming (cooling). IOD intensity is represented as the difference in SST anomalies between a western equatorial Indian Ocean (WEIO) area (50° E–70° E and 10° S–10° N) and a south eastern equatorial Indian Ocean (EEIO) area (90° E–110° E and 10° S–0° N) called the Dipole Mode Index (DMI) (Saji et al., 1999) (Fig. 1). The DMI is positive (negative) during a positive (negative) IOD event. A negative IOD event can be considered as an intensification of the normal state (presence of equatorial westerlies when EEIO is warmer than WEIO) whereas a positive IOD event represents conditions that are nearly opposite (presence of anomalous equatorial easterlies when WEIO becomes warmer than EEIO) to the normal state (Vinayachandran et al., 2009). Currie et al. (2013) found negative influence of IOD on chlorophyll concentrations along the southern tip of India during fall. They also found that within the entire Indian Ocean basin, the only region where ENSO had a greater influence than IOD on chlorophyll concentration was in the Somalia upwelling region (western Arabian Sea) where it caused a decrease in fall and



winter chlorophyll concentration by reducing the local upwelling winds. Wiggert et al. (2009) also noticed a negative chlorophyll anomaly around the southern tip of India between October and December during positive IOD events.

In addition to the inter-annual variability, SSTs in the tropical Indian Ocean show accelerated warming post the mid-1970s (Kumar et al., 2009; Nitta and Yamada, 1989; Roxy et al., 2014). Roxy et al., (2015) noted that the western Indian Ocean region is warming for more than a century, at a rate faster than any other region of the tropical oceans. Cai et al., (2014) showed that in a warming environment weakening of both equatorial westerly winds and eastward oceanic currents in association with faster warming of the western Indian Ocean leads to more frequent occurrence of extreme positive IOD events. The relationship between positive (negative) IOD and the Indian summer monsoon rainfall (ISMR) during June-September is rather complex with some studies indicating a positive (negative) influence over the Indian region (eg: Ashok et al., 2001; 2004; Ashok and Saji, 2007) whereas there are studies that have found no significant relation between IOD and ISMR (eg: Ihara et al., 2007; Gadgil et al., 2007). The variability in the onset of the Indian summer monsoon is also influenced by the phase of the IOD (Sankar et al., 2011).

Previous studies have also shown that oceanic circulation in the eastern Arabian Sea (EAS) is modulated through remote forcing by the surface winds over the equatorial Indian Ocean and alongshore winds in the Bay of Bengal (BoB) (Yu et al., 1991; McCreary et al., 1993; Shankar and Shetye, 1997; Shankar et al., 2002). Coastal Kelvin waves seen in the BoB are forced primarily by alongshore winds in the Bay or by winds in the equatorial waveguide (McCreary et al., 1993; Yu et al., 1991). They propagate eastward as equatorially trapped Kelvin waves until they reach the coast of Sumatra on the eastern boundary of the BoB, where they bifurcate into two coastally trapped Kelvin waves, one propagating northwards along the coastal wave guide of the Bay of Bengal and the other propagating southwards. In the BoB, the first upwelling (downwelling) Kelvin wave occurs during January-March (May-August) and the second upwelling (downwelling) Kelvin wave occurs during August-September (October-December) (Rao et al., 2010, Sreenivas et al., 2012). The upwelling (downwelling) Kelvin wave during the SWM(NEM) triggers the formation of an anomalous circular low (high) known as Lakshadweep low (Lakshadweep high) in the southeastern Arabian Sea to the east of the Lakshadweep islands. They are a consequence of westward propagating Rossby waves radiated by Kelvin waves off southwest India propagating poleward along the western margin of the Indian subcontinent (Shankar and Shetye, 1997; Shetye 1998). Rao et al. (2010) however, has reported that among the four Kelvin waves the second downwelling wave is the most pronounced and it alone has an influence on the circulation of the southeastern Arabian Sea. They further say that the second upwelling Kelvin wave during the summer monsoon (August-September) is least developed and limited to the south eastern rim of the Bay. They attributed this to the fact that the equatorward flowing East India Coastal Current (EICC), during October–December favors the propagation of the second downwelling Kelvin wave into the southeastern Arabian Sea. In contrast, during July-September EICC reverses in direction and flows poleward resulting in the blocking or weakening the propagation of the second upwelling Kelvin wave in the coastal wave guide of the western BoB.

The SST anomalies associated with IOD tend to appear in June, intensify in the following months and peak in the month of October followed by a rapid decline (Saji et al., 1999). June-August is considered to be the developing phase of IOD





whereas September-November is considered to be the mature phase (eg: Hong, 2008). Anoop et al., (2016) found that in the month of October different phased IOD events have significant influence on the wave climate off the central west coast of India compared to the northern and southern parts. The differential warming of the tropical oceans which can either increase or decrease the zonal temperature gradient can have severe implications on the general oceanic-atmospheric circulation

pattern (eg: Swapna et al., 2014) and accordingly marine biological activity (Roxy et al., 2016). Based on this, we hypothesize that the variability in the IOD and tropical Indian Ocean warming has influenced the surface chlorophyll concentration in the coastal waters of the EAS in the recent decades. We have accordingly studied 3 distinct regions along the western coast of India viz., area 1 (73° E-76° E, 13° N-18° N) representing the central eastern Arabian Sea (CEAS) , area 2 (74° E-77° E, 8° N-13° N) representing the south eastern Arabian Sea (SEAS) and area 3 (74° E-78° E, 5° N-8° N)

representing the southern tip of India (TIP). Our aim is to investigate the impact of the tropical Indian Ocean warming and IOD on the chlorophyll concentrations in these regions and EAS in general, for the period 1998 to 2014, especially during the month of October when IOD strength is generally at its peak.

## 2 Materials and methods

### 2.1 Data

The European Space Agency (ESA) provides through Ocean Colour Climate Change Initiative (OC-CCI, www.oceancolour.org) high quality monthly ocean color data at 4.0 km spatial resolution, available from 1998 to present (Sathyendranath et al., 2017). The OC-CCI ocean color dataset is created by band-shifting and bias-correcting different ocean color satellite sensor data from MERIS and MODIS and match these to SeaWiFS data, merging the datasets and computing per-pixel uncertainty estimates. The OC-CCI surface chlorophyll-*a* (Chl_a) concentration data from version 3.1

(Sathyendranath et al., 2018), for the period 1998 to 2014 has been used in this study. OC-CCI Chl_a data has already been verified and used to estimate the surface Chl_a concentration in the Arabian Sea by several authors (eg: Shafeeque et al., 2017; Monolisha et al., 2018; Smitha et al., 2019; Menon et al., 2019). SST data are based on extended reconstructed SST (ERSST v4) (Huang et al., 2015; Liu et al., 2015), produced on a 2°x2° grid derived from the International Comprehensive Ocean–Atmosphere Dataset (ICOADS). Monthly (monthly means of daily means) wind data at 10 m height (surface) and

850 hPa, divergence at 850 hPa at 0.25°X0.25° resolution for the period 1998 to 2014 were obtained from ECMWF (European Centre for Medium-Range Weather Forecasts) ERA-Interim global atmospheric reanalysis (Dee et al., 2011). The sub-surface water temperature data at 1°x1° resolution by Ishii et al. (2005; 2006) for the period 1998-2012 has also been used to extract the location of the 20°C isotherm. In this data, monthly objectively analysed subsurface temperature and salinity at 24 levels in the upper 1500 meters is available from 1945. The analysis is based on the World Ocean Database and

Atlas, the Global Temperature-Salinity in the tropical Pacific from the French L'Institut de recherche pour le development (IRD), and the Centennial in situ Observation Based Estimates (COBE) sea surface temperature. Monthly data of sea surface height anomaly (SSHA) at 0.25°X0.25° computed from merged data by Archiving, Validation and Interpretation of Satellite Oceanographic data (AVISO) is available at https://www.aviso.altimetry.fr/en/my-aviso.html. This data is derived using up-





to-date datasets with up to four satellites at a given time, from Jason-2/Jason-1/Envisat/Topex/Poseidon/GFO. We have used AVISO monthly SSHA data for the period 1998 to 2014 in this study. In the open ocean surface geostrophic currents are caused by the variability of sea surface elevation.

## 2.2 Methodology

Based on Aparna et al. (2012) and http://www.bom.gov.au/climate/iod, there were four positive IOD years (2006, 2007, 2008 and 2012) and two negative IOD years (1998 and 2010) during our study period. Out of the four positive years 2006 was the only year during our study period in which El Niño occurred concurrently with positive IOD (Aparna et al., 2012). 2010 was the only negative IOD year that co-occurred with the moderate intensity La Niña of 2010-11 (source: http://ggweather.com/enso/oni.htm). We have considered 2006, 2007, 2008 and 2012 as the positive IOD years and 1998 and

2010 as the negative IOD years for this analysis. Wind stress curl has been computed from the monthly zonal and meridional component of the ERA-Interim winds following Eq. (1) and Eq. (2) respectively:

$$\tau_x = \rho C_d w u \tag{1}$$

$$\tau_y = \rho C_d w v \tag{2}$$

where, $\tau_x$ = Zonal wind stress in N m$^{-2}$, $\tau_y$ = Meridional wind stress in N m$^{-2}$, $C_d$ = 1.2X10$^{-3}$ is the non-dimensional drag co-

efficient, p = 1.225 Kg m$^{-3}$ is the density of air, w = wind magnitude in m s$^{-1}$, u = zonal wind in m s$^{-1}$, v = meridional wind in m s$^{-1}$. The vertical component of the curl of the surface wind stress in N m$^{-3}$ is given by Eq. (3):

$$\frac{\delta \tau_y}{\delta_x} - \frac{\delta \tau_x}{\delta_y} \tag{3}$$

The open ocean currents are for the most part geostrophic, with the Coriolis force balancing the horizontal pressure gradient force. The surface geostrophic current vectors are calculated from the SSHA data as following Eq. (4) and Eq. (5)

respectively:

$$U_s = \frac{-g}{f} \frac{\delta \zeta}{\delta y} \tag{4}$$

$$V_s = \frac{g}{f} \frac{\delta \zeta}{\delta x} \tag{5}$$

where $U_s$ is the zonal component of the surface geostrophic current in cm s$^{-1}$, $V_s$ is the meridional component of the surface geostrophic current in cm s$^{-1}$, g = 9.8 is the acceleration due to gravity in m s$^{-2}$, f = 2 $\Omega$ sin$\varphi$ is the Coriolis parameter, where

$\Omega$ = 7.292X10$^{-5}$ is the angular speed of rotation of the earth in rad sec$^{-1}$ and $\varphi$ is the latitude.

## 3 Results

The analysis of the annual SST trend during the period 1981 to 2014 (Fig. 1) indicates that the western equatorial Indian Ocean (WEIO) is warming at a much higher rate than the eastern equatorial Indian Ocean (EEIO) region. Within the Arabian





Sea basin, the eastern basin is seen to be warming at a rate higher than the western part of the basin (Fig. 1). The average annual SSTs during this period in the EEIO area is about 0.5°C higher than that of the WEIO area (Fig. 2a and 2b). Both the WEIO and EEIO areas are getting heated up with the WEIO area warming at a rate (0.015°C per year) faster than that of the EEIO area (0.013°C per year). This is equivalent to ~0.5°C increase in SST (0.44°C for WEIO and 0.51°C for EEIO) in the

equatorial tropical Indian Ocean during 34-year period analysed here (1981-2014), post the 1970s when accelerated warming commenced in the tropical Indian Ocean. The SST trendlines of WEIO (p-value <0.001) and EEIO (p-value=0.004) are also statistically significant at the 99% and 95% confidence intervals respectively. Increase in SSTs can increase the near-surface stratification and reduce vertical mixing which in turn, will reduce the nutrient supply to the euphotic zone and inhibit primary productivity (Behrenfeld et al., 2006). All the three selected areas in the EAS show a negative trend in the Chl_a

concentrations in October, the month of our interest (Fig. 3a-3c). Among the three areas, CEAS (area 1) shows the maximum decreasing trend with a slope of -0.03 mg m$^{-3}$, whereas SEAS (area 2) and TIP (area 3) both have a mild negative slope of -0.01 mg m$^{-3}$. But the trendlines of the time series of the Chl_a concentrations at all the three areas are statistically not significant (p-values > 0.1). In addition to the decreasing trend, considerable interannual variability in the Chl_a concentration is seen in the EAS. All the three study areas show negative Chl_a anomalies during all the four positive IOD

years, with maximum reduction in areas 1 and 3 seen during the strong positive IOD event of 2012. This fact is highlighted further by the analysis of the standardized anomalies of October Chl_a concentrations at the three study areas (Fig. 4). From the figure it can be seen that all the three study areas have negative Chl_a anomalies during all the four positive IOD years. For CEAS and TIP areas, the maximum negative anomaly is seen during 2012, the strongest positive IOD during the study period. During the two negative IOD years Chl_a concentrations in CEAS and TIP show a positive anomaly whereas it is

slightly negative for SEAS. As mentioned earlier in section 1, previous studies have shown that while a negative IOD year represents an intensified normal state, it is the positive IOD year that represents anomalous conditions opposite to that of the normal state (eg: Vinayachandran et al., 2009) and hence it is the Chl_a concentrations during positive IOD years that are of particular interest for our analysis. Low (high) Chl_a concentrations seen in all the three study areas during 2000 and 2014 (2001 and 2009) are probably unrelated to the phase of IOD, as other drivers such as oceanic circulation, ecosystem

dynamics and Indian monsoon variability may also affect the anomalies in Chl_a concentrations in the tropical Indian Ocean. The surface Chl_a anomalies during positive IOD years and negative IOD years superposed by wind anomalies at 10m height (surface winds) averaged for October and September-November are presented in Fig. 5a-5b and Fig. 5c-5d respectively .The surface winds differ both in magnitude and direction during the opposite phases of IOD. Surface wind anomalies during the positive IOD years are mostly north-easterlies off the west coast of India (Fig. 5a and 5c). In the case of

negative IOD years (Fig. 5b and 5d) the wind anomalies have a strong westerly component, directed perpendicular toward the west coast of India. It is interesting to note that the Chl_a anomalies are mostly positive (respectively negative) just off CEAS and TIP during the negative (positive) IOD years. In the SEAS (area 2) a strong negative Chl_a anomaly is observed during positive IOD years but signature of a corresponding strong positive Chl_a anomaly is not observed during negative IOD years, especially near the coast. The lack of a strong positive Chl_a signature in SEAS during negative IOD years is





quite interesting and it is also observed in the time series analysis (Fig. 4). The cyclonic surface wind anomaly seen in CEAS favor upwelling whereas positive Chl_a anomalies whereas increased strength of tangential wind vectors favor upwelling off the southern tip of India (TIP) during negative IOD years (Fig. 5d). In SEAS the wind anomalies are more or less perpendicular to the Indian coast and hence upwelling is not so strong during the negative IOD years (Fig. 5b and 5d).

The DMI index based on ERSST SST data are calculated as defined in section 1 for the period 1998-2014, averaged for the months September to November, and the linear correlation coefficients with spatial Chl_a concentrations (September-November) in the EAS are shown in Fig. 6a. The panel on the right (Fig. 6b) highlights those regions with positive (solid red color) and negative (solid blue color) correlations significant at the 95% confidence interval or above. In general, there is a positive correlation between the open ocean DMI and Chl_a concentration in the western Arabian Sea, whereas in the north-
eastern Arabian Sea (between 18° N and 24° N) there are only few regions with strong (and significant) positive correlations. The correlations are by and large negative, but not statistically significant in CEAS and SEAS whereas in TIP they are negatively significantly correlated. The correlations between September-November DMI and December-February Chl_a concentrations are also shown to ascertain the influence of fall IOD on the ensuing and possible inhibition of winter blooms (Fig. 6c). The right panel (Fig. 6d) highlights regions with positive (solid red color) and negative (solid blue color)
correlations significant at the 95% confidence interval or above. September-November DMI seems to have a stronger influence on the winter blooms, with almost the entire southern Arabian Sea basin having Chl_a concentrations significantly negatively correlated. Though a narrow region of positive correlations still exists along the Indian west coast they are not statistically significant. An increase in DMI accounts for a warmer than normal Arabian Sea and deeper thermocline and resultant decrease in the zonal SST gradient in the equatorial Indian Ocean. Such a scenario has adversely affected the
surface Chl_a concentration in the EAS during fall and more strongly in winter south of 12° N (Fig. 6c and 6d), though its impact on the near-shore Chl_a concentration is not conclusive (statistically not significant).

Surface winds (winds at 10 m height) and SSTs averaged for October during positive IOD years and negative IOD years are shown in Fig. 7a and Fig. 7b respectively. During positive IOD years the surface winds are weak in the EAS with a north to north-easterly component and a weak cyclonic circulation. Weak winds coupled with high SSTs in the EAS can inhibit
coastal upwelling. During negative IOD years surface winds are strong north to north-westerlies with wind vectors converging strongly over the TIP area. A study by Anoop et al. (2016) had also found that the winds are north easterlies over the central west coast of India during the positive IOD years and as such they do not support coastal upwelling when compared to the negative IOD years. Lower SSTs and strong cyclonic surface winds can induce stronger than normal upwelling along CEAS as also evidenced by the higher than normal Chl_a concentrations during 1998 and 2010 (Fig. 3a).
Despite October being a monsoon transition month, this low-level atmospheric circulation is quite deep and is also seen at the 850 hPa level. The average circulation at 850 hPa level overlaid with divergence at 850 hPa valid for the month of October during negative IOD and positive IOD years are depicted in Fig. 7c and Fig. 7d respectively. Strong cyclonic north-westerlies are seen extending till 850hPa during the negative IOD years. Negative divergence (convergence) is seen over the TIP area on account of strong tangential winds off the southern tip of India. The strong divergence parallel to the south west



coast covering most of CEAS and SEAS during the negative IOD years indicates the interaction of 850hPa wind with elevated land-mass (Western Ghats in southern India).

The shear stress exerted by winds on the surface of open ocean is represented in terms of the magnitude of the surface wind stress vectors. The vertical component of the curl of the surface wind stress can be used as an indicator of the Ekman mass

transport with positive (respectively negative) wind stress curl implying net surface divergence and upwelling (respectively convergence and downwelling) and hence cooler (warmer) SSTs. Wind stress curl is mostly positive over the EAS during the positive IOD and negative IOD years, but it is considerably stronger during the negative IOD years (Fig. 8a and 8b). Strong positive wind stress curl coincides with regions where wind stress vectors converge during the negative IOD years. There is a strong (respectively weak) cyclonic circulation in the EAS with strong (weak) convergence of wind stress vectors in the

south west coast during negative (positive) IOD years. This means are conditions are much more conductive to coastal upwelling in the EAS during negative IOD years. The June-August (summer monsoon) surface wind stress vectors superposed with wind stress curl averaged for the positive IOD and negative IOD years are shown in Fig. 9a and Fig. 9b respectively. Positive wind stress curl can be seen north of the axis of the Findlater jet (at ~18° N, 64° E indicating upwelling (Muraleedharan and Prasannakumar, 1996). Wind stress curl remains positive all along the west coast of India during

positive and negative IOD years, but the strength and offshore extent are lower during positive IOD years. This means that when even positive IOD is in its developing phase (during June-August), conditions have become decreasingly favorable to induce coastal upwelling along the entire west coast of India.

The depth of the thermocline is an important factor that determines the extent of upwelling, with shoaling (respectively deepening) of thermocline associated with upwelling and surface divergence (downwelling and surface convergence). The

20℃ isotherm is typically located near the center of the main thermocline in tropical waters. As a result the depth of the 20℃ isotherm from the surface, known as D20 has been broadly used as a proxy for thermocline anomalies (eg: Meyers, 1979; Vialard and Delecluse, 1998). The upliftment (respectively deepening) of D20 can be considered as an upwelling (downwelling) signature. The October month D20 averaged for the positive IOD years (Fig. 8c) and negative IOD years (Fig. 8d) highlights the difference in thermocline depth off the west coast of India. D20 shoals to approximately 50 m near

south eastern Arabian Sea in the negative IOD years, whereas it is close to 80 m during the positive IOD years. Along the entire west coast of India, sub-surface conditions are more conductive for wind induced coastal upwelling during the negative IOD years. The June-August (SWM) D20 averaged for the positive IOD years (Fig. 9c) and negative IOD years (Fig. 9d) identifies two deep MLD regions in the central Arabian Sea separated by the axis of the Findlater jet. The deep MLD to the north of the Findlater jet during negative IOD years indicate strong wind mixing and positive wind stress curl.

The signature of increased D20 shoaling along SEAS during negative IOD years originates with the onset of summer monsoon that coincides with the developmental phase of a typical positive IOD.

We have analysed the mean SSHA derived from aviso and the geostrophic currents calculated from SSHA during October averaged for positive IOD (Fig. 10a) and negative IOD years (Fig. 10b). The surface currents off the west coast of India are directed equatorward known as the west India coastal current (WICC) during SWM. The westward propagating Rossby





waves that originates off the south west coast of India as the Lakshadweep low during the SWM has by October, spread all over the southern Arabian Sea. The surface geostrophic currents are cyclonic around this low, the circulation being more intense during the negative IOD years. In the monsoon transition month of October the WICC reverses in direction and flow poleward during October-January. The negative sea level anomaly is also more pronounced during the negative IOD years

and supports stronger coastal upwelling along CEAS and SEAS. The signal of downwelling Kelvin waves generated by the equatorial westerlies during October-December (Rao et al., 2010) can be seen as a positive SSHA in the south east coast of India, turning around Sri Lanka and entering the south west coast of India. The influence of the downwelling Kelvin waves become more apparent in December (Fig. 10c and Fig. 10d) because of the time taken by the wave in traversing the entire Bay of Bengal boundary and around Sri Lanka to reach the Arabian Sea. Shetye (1998) had reported that the coastally

trapped Kelvin waves originating from the Sumatra coast move along the periphery of BoB and reach the northern tip of Sri Lanka in about 28 days. The downwelling Kelvin waves generate a positive SSHA off the south west coast of India with anti-cyclonic geostrophic currents, replacing the Lakshadweep low that was present during summer monsoon. The weak positive SSHA seen during positive IOD years (Fig. 10c) result in weak geostrophic currents along the entire Arabian Sea and also have a negative impact on the strength of the poleward flowing WICC. In the case of negative IOD years

conditions favoring upwelling are seen in the open ocean, to the west of 72° E whereas during positive IOD years the cyclonic geostrophic currents associated with negative SSHA are very weak to trigger upwelling. This could be the reason for the statistically significant anticorrelation between DMI and winter Chl_a concentration within the region 60° E-78° E; 5° N-12° N as seen in Fig. 6d.

**4 Conclusions**

The influence of increased warming and positive IOD on the marine primary productivity of the EAS during boreal fall has been studied using various data records of biogeochemical ocean and atmospheric variables during the period 1998-2014. Post the mid-1970s, the WEIO has warmed at rate higher than EEIO thereby reducing the zonal gradient of SST in the equatorial Indian Ocean. This decreasing trend in zonal gradient has altered the tropical Indian Ocean circulation to a great extent as also shown by Roxy et al., (2014). As a result of this asymmetrical warming, the conditions favoring the formation

of positive IOD years are becoming more frequent when compared to those favoring negative IOD years. Within the Arabian Sea the EAS shows a higher warming trend when compared to that of the western basin. During positive IOD years also the EAS remains warmer than the western part of the basin. Higher SSTs in the EAS result in a deeper thermocline thereby suppressing the supply of cool nutrient-rich water from the sub-surface. All the three study areas in the EAS show a steady decreasing trend in the surface Chl_a concentrations, though statistically not significant. The surface winds over CEAS and

SEAS are cyclonic and north to north-westerly during positive IOD years. These strong winds parallel to the coast (alongshore winds) induce coastal upwelling in these regions (Joseph et al., 2018). During positive IOD years the surface winds are weaker with a weak cyclonic circulation and north to north-easterly in direction. The winds are very strong with strong convergence of wind vectors in TIP area during negative IOD years. Anomalous surface winds tangential to the





southern tip of India trigger anomalous positive concentrations of surface Chl_a. These results are in accordance with the previous studies by Currie et al. (2009) and Wiggert et al. (2013), who had reported a negative chlorophyll anomaly during positive IOD years along the south west coast and off the southern tip of India during fall.

Observational study by Gopalakrishna et al. (2008) had shown that for the formation of Lakshadweep low during the SWM, alongshore winds (local winds) off the southwest coast of India combined with remote forcing from south of Sri Lanka are far more important when compared to remote forcing from the equatorial winds. Our analysis has shown that local surface winds in the EAS are less favorable for upwelling during positive IOD years, both in terms of magnitude and direction. Positive IOD years are characterized by the presence of extended easterly anomalies (generating upwelling Kelvin waves) along the equator. As explained in Sect. 1, the influence of upwelling Kelvin waves generated by the equatorial waveguide is negligible in the southeastern Arabian Sea during SWM due to the presence of the poleward flowing EICC. This results in the formation of a weak Lakshadweep low during SWM. The propagation of this weak low westward as a Rossby wave initiates less intense upwelling in the entire southern Arabian Sea. During October-December with the EICC flowing equatorward, downwelling Kelvin waves reach the southeastern Arabian Sea. However, during positive IOD years, due to presence of extended easterly wind anomalies, the strength of the downwelling currents are considerably low. Similarly, during the negative IOD years, strong local alongshore winds help in the development of a strong Lakshadweep low in the southeastern Arabian Sea during SWM. During October-December, the presence of stronger than normal equatorial westerly wind anomalies generate stronger downwelling Kelvin waves that propagate around the periphery of BoB and reaches the southeastern Arabian Sea. The nest result is the formation of a weak (accordingly strong) cyclonic low in the southeastern Arabian Sea during SWM and a weak (strong) geostrophic current flowing into the southeastern Arabian from the BoB and further advancing poleward as the WICC during October-December in the case of positive (negative) IOD years. The correlation analysis between DMI index and Chl_a concentration in the EAS shows that open ocean Chl_a concentration (from ~60° E to the west coast of India, southern tip of the subcontinent to 12° N) is mostly significantly negatively correlated with the DMI index. This means that in the event of the occurrence of frequent positive IOD years in the future, which is more likely the scenario with the WEIO warming more than EEIO, the surface Chl_a concentration during fall is likely to decrease.

**Author contributions**

Conceptualization of this paper was done by SS, ATR, KGFE, DK, RS, RM and LHP. The methodology, validation, formal analysis and software were done with contributions from SS, ATR, KGFE, DK, and RS. Writing—original draft preparation was carried out by SS with contribution from all co-authors. Writing—review and editing was done by SS, ATR, KGFE, DK, RS, RM and LHP. Supervision of the paper work and funding acquisition for publication were done by LHP.

**Competing interests**

The authors declare that they have no conflict of interest.



**Acknowledgements**

This paper is the outcome of the study carried out during the International Winter School on "Operational oceanography: Indian Ocean circulation and sea level variation" jointly organized at International Training Centre for Operational Oceanography (ITCOocean) in Hyderabad, in cooperation with Nansen Environmental and Remote Sensing Center (NERSC), Norway, Nansen Scientific Society (NSS), Norway, Nansen Environmental Research Institute-India (NERCI), Cochin and Indian National Centre for Ocean Information Services (INCOIS). SS acknowledges the financial support from NERSC, Norway. ATR and RM acknowledge the help from director of ESSO-NCESS for providing encouragement to carry out this work. KGFE thanks NERSC, NSS, NERCI and INCOIS for providing the opportunity to attend the winter school. RS is thankful to Prof. Arun Chakraborty, IIT Kharagpur for providing the opportunity to attend the winter school. The authors are grateful to Dr. N. Nandini Menon, NERCI for valuable comments and suggestions.

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



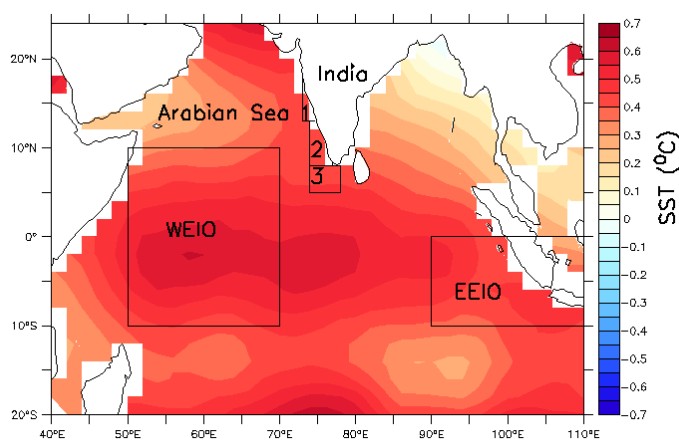

**Figure 1: The map of tropical Indian Ocean highlighting the study area. WEIO represents the western equatorial Indian Ocean (50° E–70° E , 10° S–10° N, EEIO represents the eastern equatorial Indian Ocean (90° E–110° E, 10° S–0° N), area 1 (73° E-76° E, 13° N-18° N) represents the central eastern Arabian Sea, area 2 (74° E-77° E, 8° N-13° N) represents the south eastern Arabian Sea and area 3 (74° E-78° E, 5° N-8° N) represents the southern tip of India. The annual SST trend (°C) during the period 1981 to 2014 is also shown in the image.**

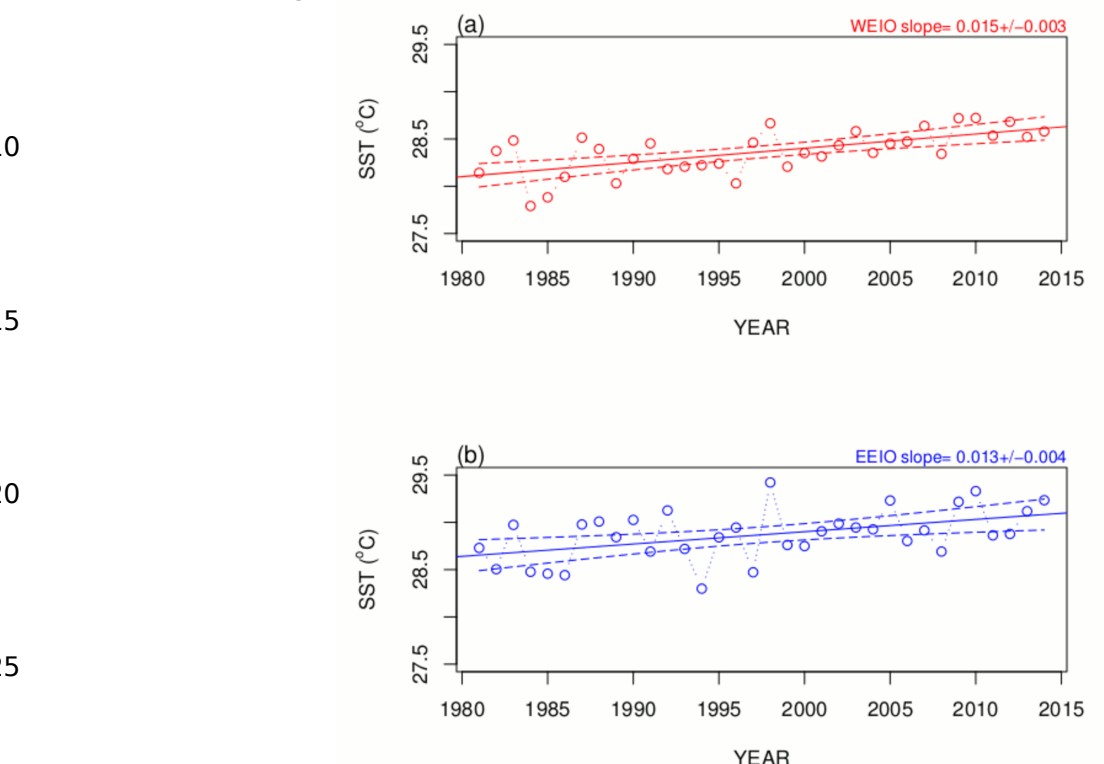

**Figure 2: Time series of annual SST (in °C) is shown for (a) WEIO and (b) EEIO using circles for annual values joined by dotted lines for the period 1981 to 2014. The solid line represents the trendline of the time series whereas the dashed lines represent the**





95% confidence interval around the trendline. The slope of the trendline +/- standard error along with p-value is shown on the top of each figure.

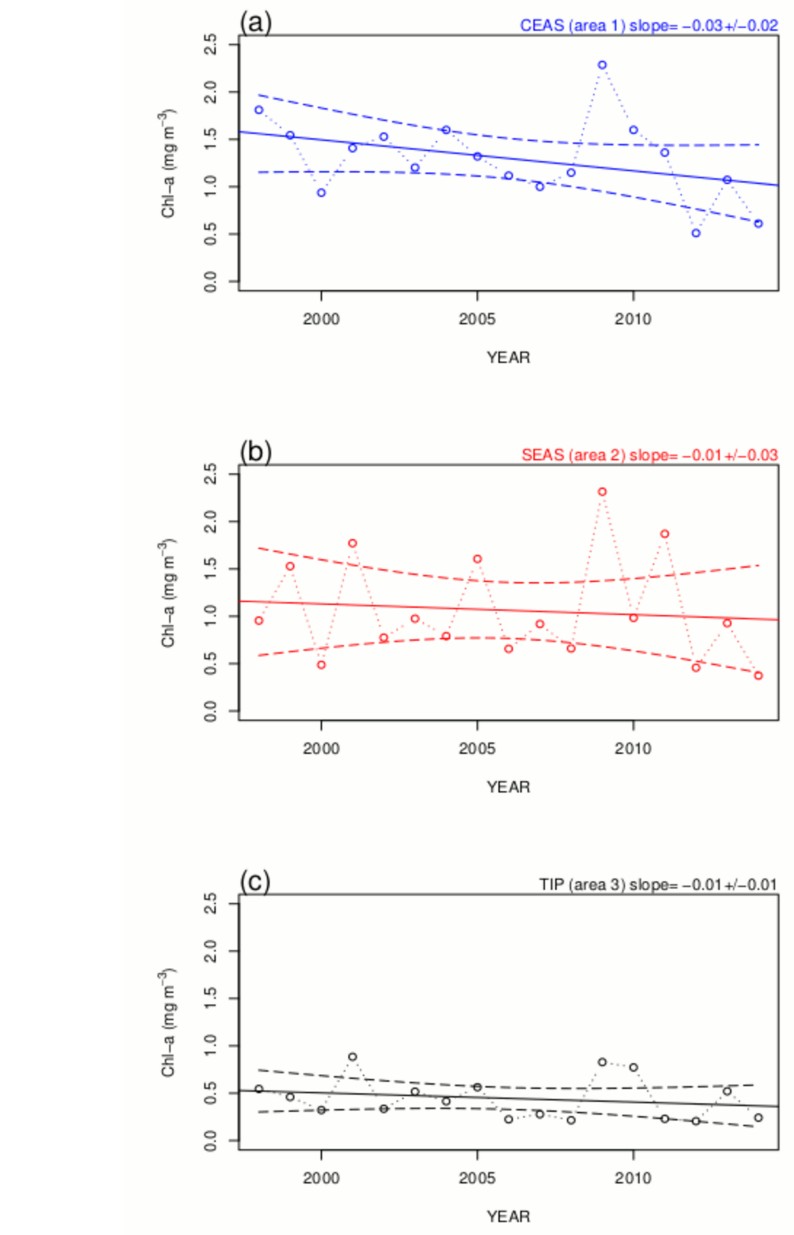

**Figure 3:** The time series of surface Chl_a concentrations (in mg m$^{-3}$) averaged for the month of October plotted for (a) CEAS (area 1), (b) SEAS (area 2) and (c) for TIP (area 3) using circles joined by dotted lines for the period 1998 to 2014. The solid line represents the trendline of the time series whereas the dashed lines represent the 95% confidence interval around the trendline. The slope of the trend line +/- standard error along with p-value is shown on the top each figure.





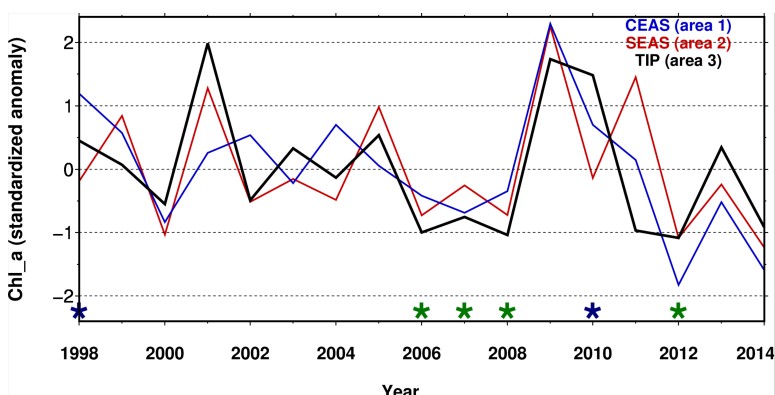

**Figure 4: The standardized anomaly of the surface Chl_a concentrations averaged for the month of October is plotted for (a) CEAS (area 1), (b) SEAS (area 2) and (c) for TIP (area 3). The green stars represent the positive IOD years and blue stars represent the negative IOD years during the study period.**

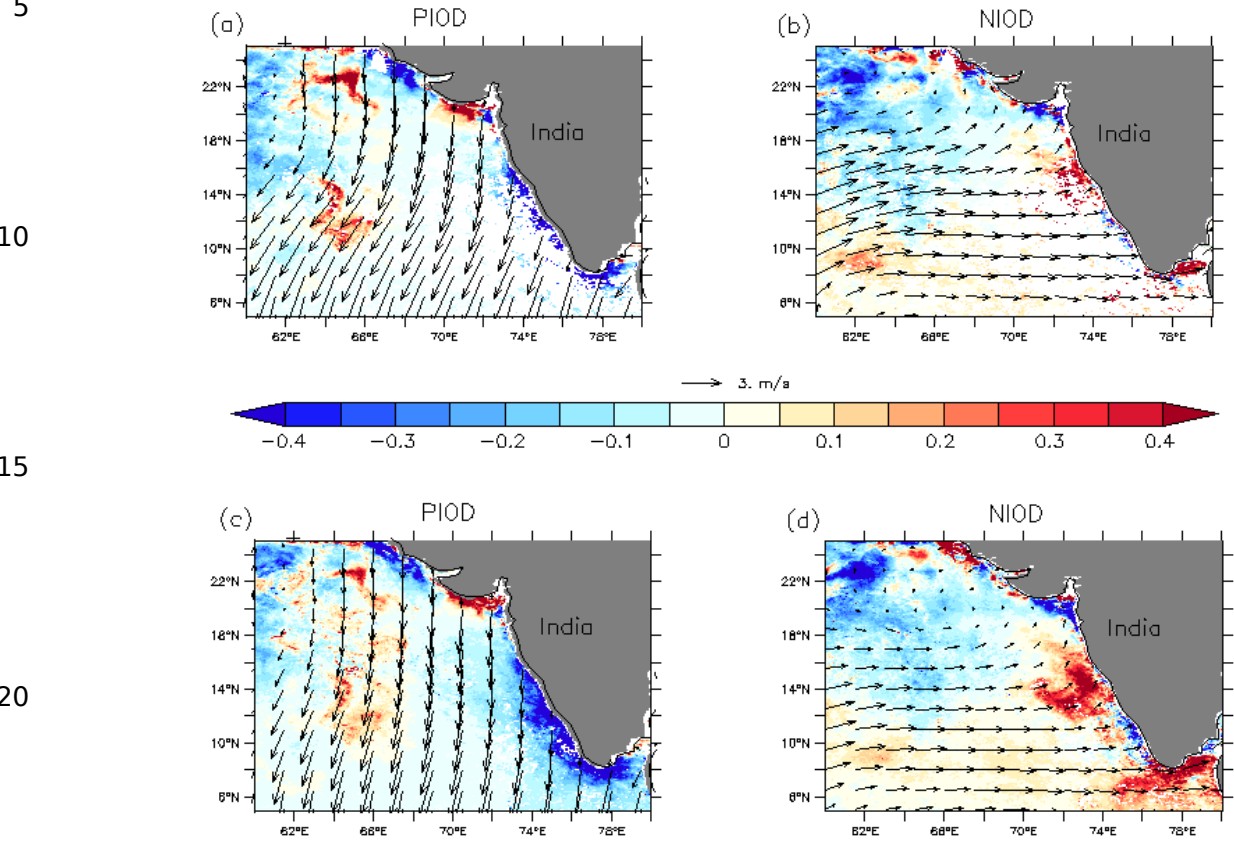

**Figure 5: Chl_a anomalies (in mg m$^{-3}$ according to color bar) overlaid with wind vector anomalies at 10 m height (surface wind in m s$^{-1}$) valid for (a) month of October during positive IOD years (b) October during negative IOD years (c) September-November during positive IOD years and (d) September-November during negative IOD years.**


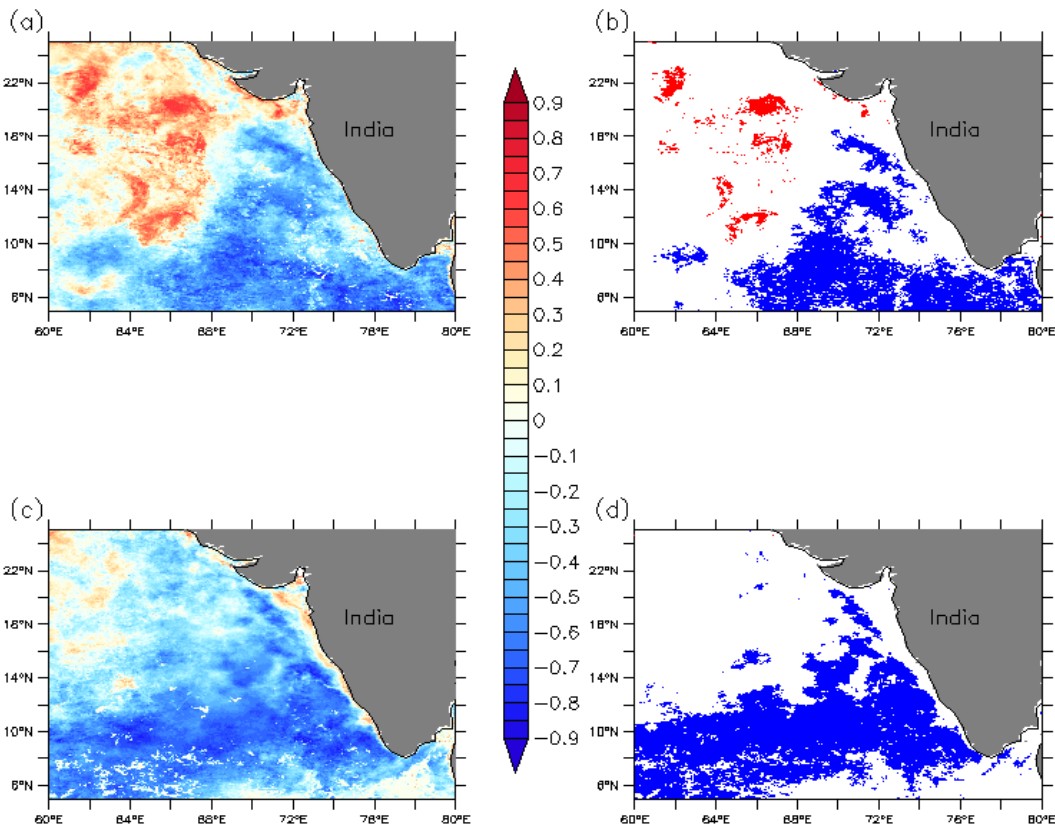

**Figure 6: (a) Correlations between September-November DMI index and September-November Chl_a concentrations in the eastern Arabian Sea for the period 1998-2014. (b) Same as that of (a) but only those regions are highlighted where the positive (solid red color) and negative (solid blue color) correlations are significant at the 95% confidence level or more. (c) The correlations between September-November DMI index and December-February Chl_a concentrations in the eastern Arabian Sea for the period 1998-2014. (d) Same as that of (c) but only those regions are highlighted where the positive (solid red color) and negative (solid blue color) correlations are significant at 95% confidence level or more.**

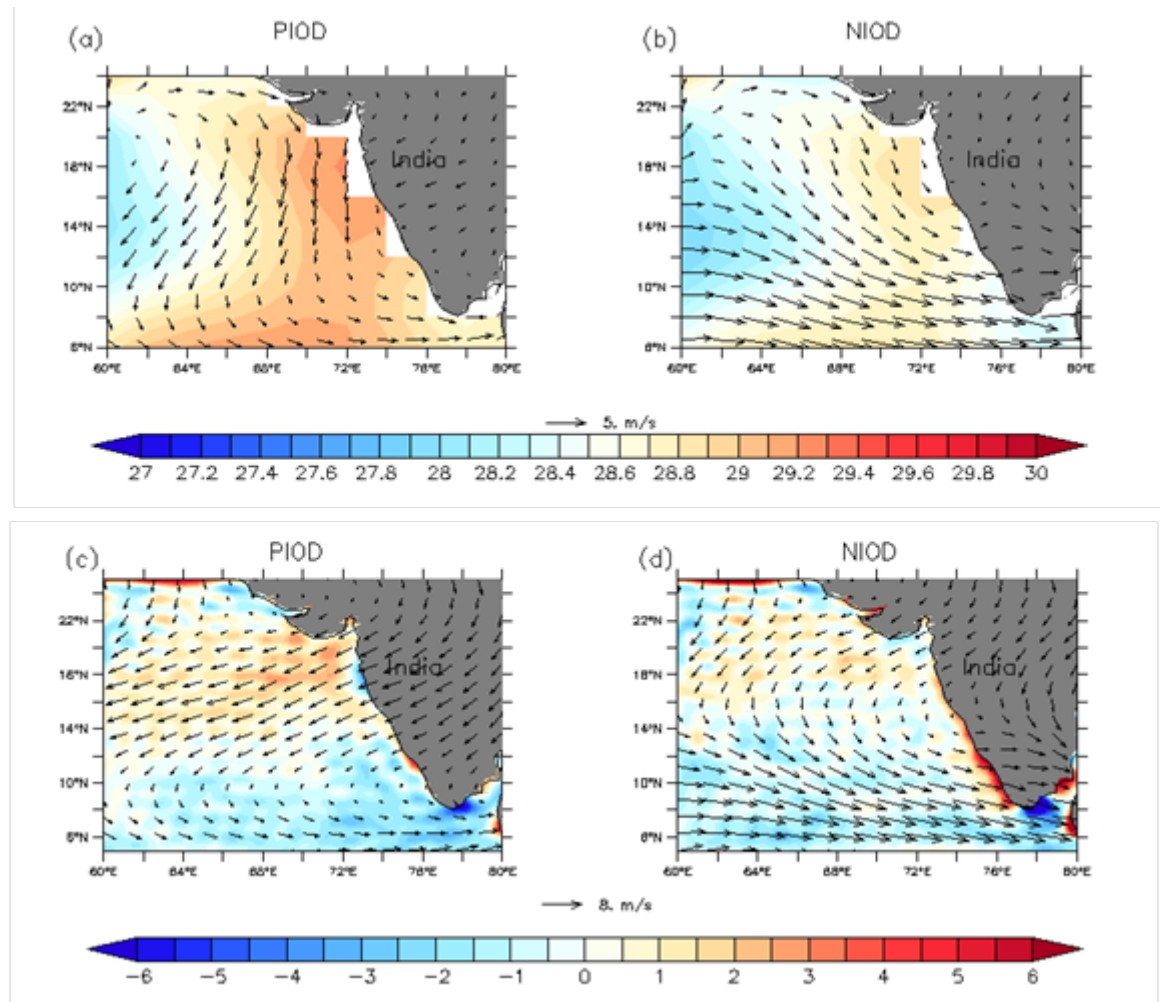

**Figure 7: The 10 m height mean wind vectors (surface wind in m s⁻¹) overlaid with mean SSTs (ºC) valid for October during (a) positive IOD years (b) negative IOD years (c) The 850 hPa wind vectors (m s⁻¹) overlaid with divergence X (10⁻⁶ s⁻² ) at 850 hPa valid for October during positive IOD years (d) negative IOD years.**





**Figure 8: Surface wind stress vectors (N m$^{-2}$) overlaid with wind stress curl in 10$^{-7}$ N m$^{-3}$ valid for (a) October during positive IOD years and (b) October during negative IOD years (c) The depth of D20 (m) valid for October during positive IOD years and (d) negative IOD years.**





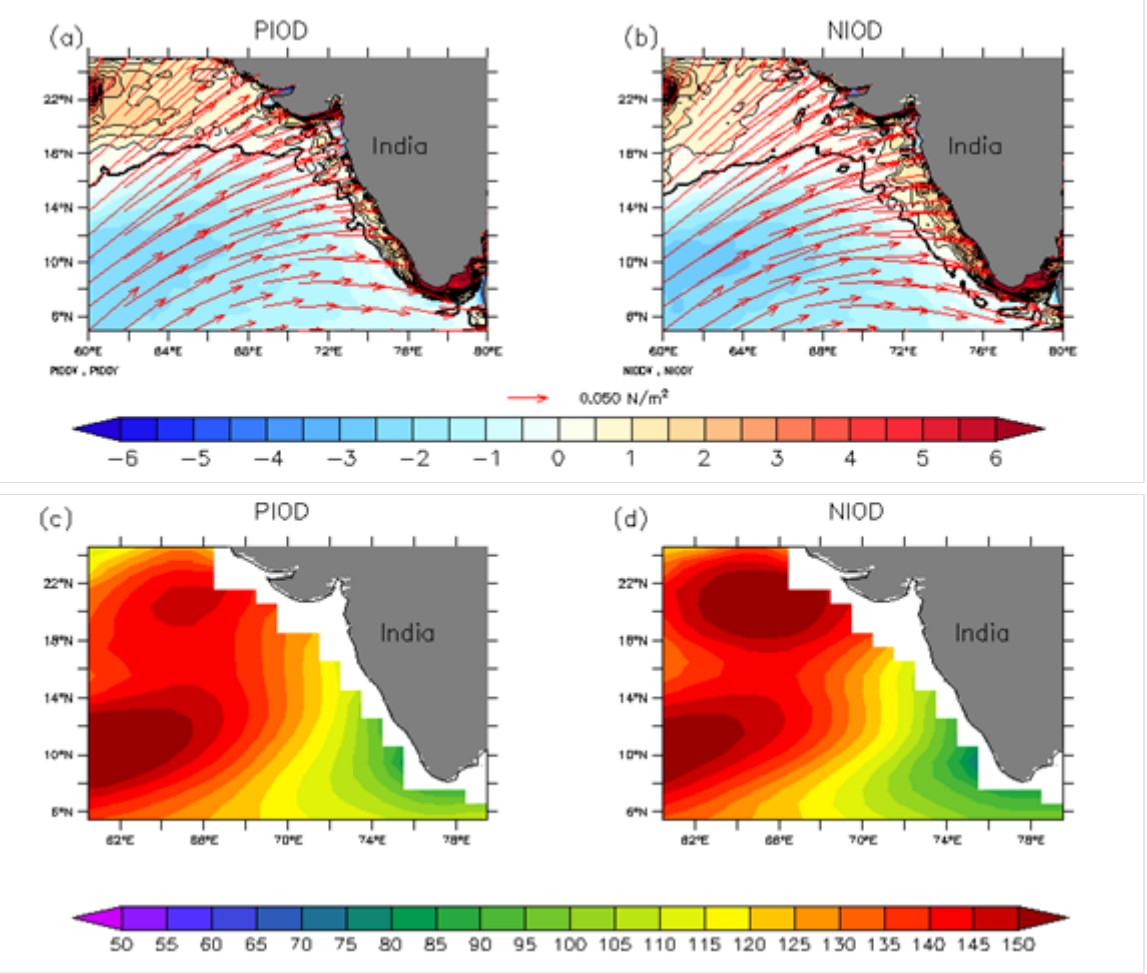

**Figure 9: Surface wind stress vectors (N m$^{-2}$) overlaid with wind stress curl in 10$^{-7}$ N m$^{-3}$ valid for (a) June-August during positive IOD years and (b) June-August during negative IOD years (c) The depth of D20 (m) valid for June-August during positive IOD years and (d) negative IOD years.**



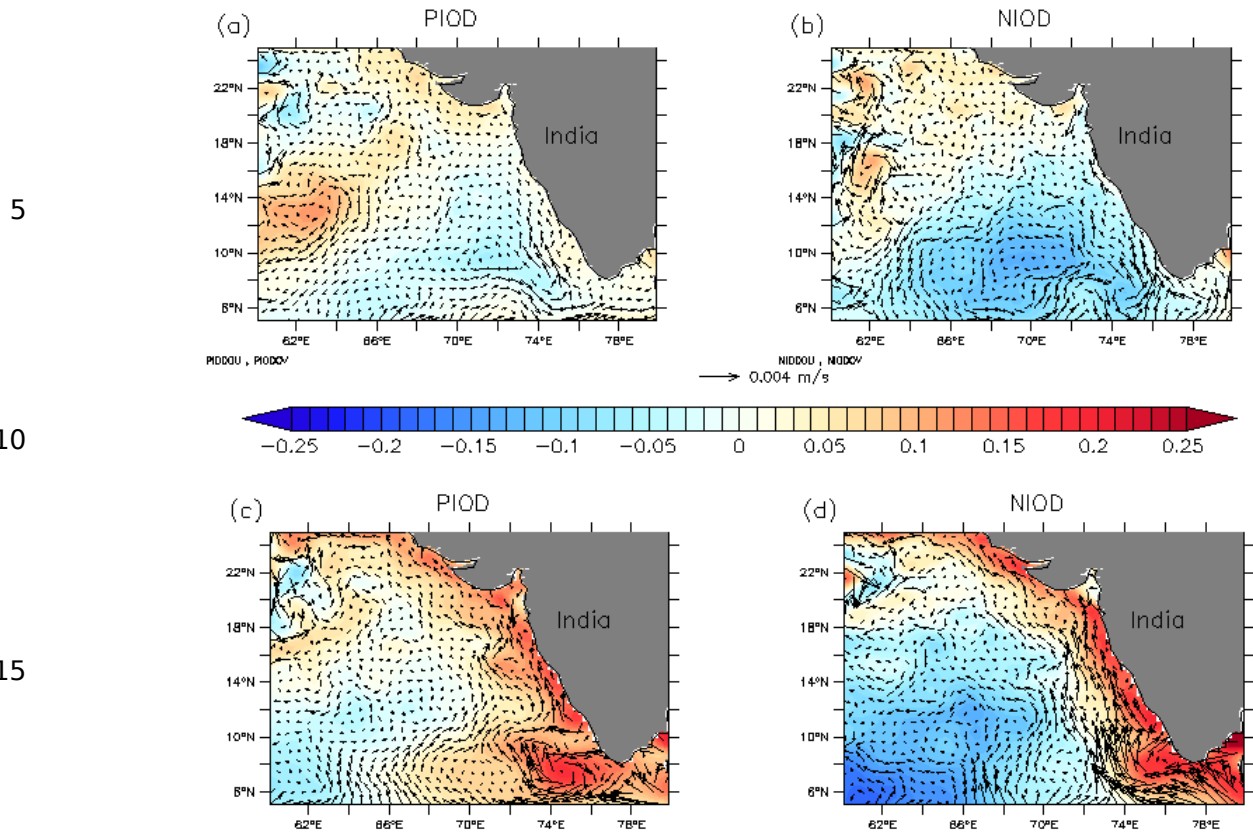

**Figure 10: SSHA (m) overlaid with surface geostrophic current vectors (m s⁻¹) valid for (a) October during positive IOD years (b) October during negative IOD years (c) December during positive IOD years (d) December during negative IOD years.**