# Peer review of "The influence of tropical Indian Ocean warming and Indian Ocean Dipole on the surface chlorophyll concentration in the eastern Arabian Sea"

_Biogeosciences, 2019_

## Referee Comment (RC1) · Anonymous Referee #1 · 24 Jun 2019

The tropical Indian Ocean warming and Indian Ocean Dipole on the surface chlorophyll concentration in the eastern Arabian Sea is an important topic. This paper shows that the increased IO warming and positive IOD events causes the decreasing trend of Chla in the southeastern AS. But earlier studies ( for example, Currie et al., 2013; Wiggert et al., 2009, 2006; Murtugudde et al., 1999 ) documented about the role of IOD on the chlorophyll variability in the EAS especially during fall. Their study also discussed the reason for Chla variability during the IOD events. In this context, how your study is different from their study. You need to justify it. Explain the mechanism and processes which lead to the asymmetrical warming, and how it favors the formation of frequent positive IOD years. Content comments Abstract P1, line 15: Why you limited your

study period up to 2014 when chlorophyll data is available up to the end of 2018 ? P1, lines 22-15: How you can say that the occurrence of positive IOD years under a global warming regime affect the chlorophylla trend. To confirm that find the rate of warming and associated chlorophyll variations when the IOD events are absent and compare it with the period when IOD events are frequent. I believe that the decreasing trend of chlorophyll is more related to the increasing water column stratification rather than the IOD events; however, inter-annual variability might be contributed by the IOD events. Data and Methodology You are discussing about IOD and El Nino–Southern Oscillation in the Introduction, but ENSO contribution towards the Chla variability is not addressed in the ms. Why your study is restricted only up to 2014 ? P 5, lines 15-23: replace "p" with "rho" . Similarly, symbol used for wind curl and geostrophic current is also wrong. Please use correct symbol. Result P5, line 27: What basis you have taken SST trends from 1981 onwards; not before 1981; and why it is restricted up to 2014 when data is available up to the recent year? P 6, line 18 : What basis you said that 2012 is the strongest IOD year during the study period. I feel 2006 is the strongest. Cross check it with DMI index. P7, line lines 1-2 : Sentence is not correct. P8, line 20: Can you show time series of D20 in your boxes ? P8, line 10: conductive ? I think wrongly written Conclusion In your conclusion you have mentioned that asymmetrical warming favoring the formation of frequent positive IOD years. Is it your result ? If so, I have not seen this part discussed elsewhere in the ms. Which is very important. Why EAS shows higher warming than the western AS ? Can you show an SST trend of western AS in the same period ? Page 10, line 18: is it nest ? Figure 3. When Chla trend in all the three boxes are not sginificant, why the study is important?

---

## Referee Comment (RC2) · Anonymous Referee #2 · 2 Jul 2019

General Comments:

The authors use remotely sensed chlorophyll pigment concentration data from OC-CCI for the period 1998 to 2014 to study the surface chlorophyll variability in 3 small boxes (very close to the coast in the latitude bound of 5-8oN & longitude bound of 73-78oE) in the eastern boundary of the Arabian Sea in the context of warming Indian Ocean and Indian Ocean Dipole. This subject was dealt with several earlier researchers and we do have some mechanistic understand of rapid warming of tropical Indian Ocean and its impact on chlorophyll. See for example, Goes et al., 2005; Prakash and Ramesh, 2007; Levy et al., 2007; Prasanna Kumar et al., 2010; Roxy et al., 2016, some of which the

authors have cited in their ms. Role of IOD has also been explored by several authors to understand the interannual variability of chlorophyll in the Arabian Sea/Indian Ocean (see for example, Currie et al., 2009; Wiggert et al., 2013).

The only two results emanating from the study is that (1) in the 3 small boxes that the authors have chose to study chlorophyll during 1998 to 2014, a steady decreasing trend in the surface chlorophyll concentration was noticed which did not show any statistical significance, (2) under warming Indian Ocean and frequent occurrence of positive IOD the surface chlorophyll concentration is likely to decrease. The later is already reported by earlier authors giving reasons for such occurrence. However, in the present study authors do not provide any mechanistic relationship to explain why chlorophyll should decrease. Major concerns:

• My major concern is that I do not understand what the motivation and purpose of the present study is. Neither do I understand what the authors want to convey to the reader.

• In the 11 lines of abstract there is no substantial result that is worth reporting. As indicated under general comment the first result is inconsequential as the decreasing trend in surface chlorophyll is not statistically significant. The 2nd result is already known as previous authors have already reported declining trend in the chlorophyll concentration using long term data as well as model simulation (see for example Roxy et al., 2016; Currie et al., 2009 and Wiggert et al., 2013).

• Why the authors have chosen only 3 tiny boxes close to the coast along the eastern boundary to examine the role of IOD.

• October is a transition month from summer monsoon to winter monsoon. What is the rationale in selecting only the month of October for the study? Why not consider the coastal upwelling months of June to September?

• What is the role of EICC in bringing about the variability in chlorophyll concentrations?

• How IOD, surface chlorophyll and Kelvin wave is connected? It is not clear from the manuscript.

• Why study period was restricted to 2014 from 1998, when the chlorophyll data is available until 2018?

• In the study area (2 tiny boxes along the eastern boundary) data on D20 is mostly not available as could be seen from Figs. 8 & 9. In such case how reliable are the inferences that the authors draw based on these diagrams?

• How the asymmetric warming favours formation of more frequent formation of positive IOD?

• What is/are the reason/s for the higher warming of the eastern Arabian Sea compared to the western Arabian Sea? This is not explained anywhere in the ms.

1. If the steady declining trend in surface chlorophyll in all the 3 small boxes along the eastern Arabian Sea is statistically insignificant

Minor concerns:

• What is the purpose of lines 13 to 32 in page 3? It can be deleted as it do not contribute to the theme of the ms.

• What is "indigenous" IOD (line 19, page 2)

• What is the "marine biological activity"? (line 5, page 4)

• In equations 3, 4 & 5, the notations for curl, zonal and meridional component of geostrophic current respectively are incorrect.

• What is "standardized anomalies" (line 16, page 6)

• Lines 16-20, Page 6: The authors cannot brush aside this by simply saying that our nterest is only in positive IOD. The authors need to address "Why Seas chlorophyll

anomaly alone showed a negative value while others were positive"

• Lines 23-25, Page 6: What is the basis for this statement? If it is so why such drivers are not being considered during positive phase of IOD?

• Line 30, Page 7: How this is possible? During transition the winds over the Arabian Sea is weak and variable.

• Line 4, Page 8: What is "vertical component"? This is not Ekamn mass transport. Authors may like to consult standard Oceanographic text book.

• Lines 22-23, page 8: Upwelling is not the only process that brings about the up-sloping and down-sloping of isotherms. Modify.

• Lines 5-10, Page 9: I cannot understand what authors wish to communicate? This is well documented by previous authors. How this is relevant in the present context.

• Lines 12-14, page 9: This is ambiguous. Authors need to provide a robust mechanism that unambiguously addresses how WICC would impact change in chlorophyll.

―――――――――――――――――――――――――

---

## Author Comment (AC1) · 9 Aug 2019

Reply to reviewer 1 We would like to thank the anonymous reviewer for the valuable comments and suggestions. The reply to reviewer's queries are listed below. A pdf version of the reply including the figures is also attached herewith.

In this context, how your study is different from their study. You need to justify it. Explain the mechanism and processes which lead to the asymmetrical warming, and how it favors the formation of frequent positive IOD years. We have tried to address the chlorophyll trend for the period 1998 to 2018 during the Sep-Nov months when IOD is in its mature phase. We found that the chlorophyll trend in the CEAS box shows

a decreasing trend that is statistically significant at the 95% confidence level. In the SEAS box, however there no statistically significant trend (Fig. 1). Close to the coast in the SEAS there are regions with positive chlorophyll trends whereas away from the coast chlorophyll mostly shows a decreasing trend. The increased warming trend in the eastern Arabian Sea is a major factor to influence the negative chlorophyll trend in the area CEAS. The chlorophyll concentration close to the coast in the SEAS is greatly influenced by local as well as remotely forced winds. Further studies are needed to ascertain the reasons for the increasing trend of chlorophyll concentration just off the south west coast of India and along the tip of the subcontinent. We have modified the text accordingly in the revised manuscript. We meant asymmetrical warming in the equatorial Indian Ocean region. The western equatorial Indian ocean region is warming at a rate faster than the eastern equatorial Indian Ocean. The increased warming of the colder western equatorial Indian Ocean (when compared to the eastern equatorial Indian Ocean) may result in the increased frequency of formation of IOD years. This has been reported by Roxy et al. (2014) in their study. Fig. 1 Chlorophyll trend for the Sep-Nov months durring the period 1981-2018. area 1 (65° E-75° E 12° N-18° N) represents the central eastern Arabian Sea (CEAS), area 2 (65° E-76° E, 7° N-12° N) represents the south eastern Arabian Sea (SEAS).

Abstract P1, line 15: Why you limited your study period up to 2014 when chlorophyll data is available up to the end of 2018 ? We have now extended our analysis till 2018 making use of OC-CCI chlorophyll data version 4. In the previous manuscript we had used OC-CCI chlorophyll data version 3.1 which does not cover 2018 entirely.

P1, lines 22-15: How you can say that the occurrence of positive IOD years under a global warming regime affect the chlorophylla trend The DMI index is significantly and negatively correlated with the surface chlorophyll concentration during fall. In the event of occurrence of frequent positive IOD years under a global warming regime, the surface chlorophyll concentration is likely to decrease during fall.

To confirm that find the rate of warming and associated chlorophyll variations when

the IOD events are absent and compare it with the period when IOD events are frequent. We have computed the difference between positive IOD and negative IOD years (left panel Fig. 2) and the difference between positive IOD and IOD neutral years (right panel, Fig. 2). Negative chlorophyll concentration is clearly visible along the entire west coast when compared to that of neutral IOD years. From this figure it is clear that during the positive IOD years when the western equatorial Indian Ocean is warmer than its eastern counterpart chlophyll shows a negative anomaly. This is due to the increased stratification and reduced vertical mixing associated with the increased SSTs (Behrenfeld et al., 2006). Coastally trapped Kelvin waves generated by equatorial wind anomalies also influence the thermocline depth along the west coast of India (Parvathi et al., 2017). Data and Methodology You are discussing about IOD and El Nino–Southern Oscillation in the Introduction, but ENSO contribution towards the Chla variability is not addressed in the ms. We have mentioned that the influence of IOD on the chlorophyll concentration is higher when compared to that of El Nino on the eastern Arabian Sea (Currie et al., 2013). We have not separated out the influence of El Nino and IOD in our study.

Why your study is restricted only up to 2014 ? We have extended our study till 2018.

P 5, lines 15-23: replace "p" with "rho" Modified appropriately in the revised manuscript.

Similarly, symbol used for wind curl and geostrophic current is also wrong. Please use correct symbol. We have now corrected the symbols of wind curl and geostrophic current in the modified manuscript.

Result P5, line 27: What basis you have taken SST trends from 1981 onwards; not before 1981; and why it is restricted up to 2014 when data is available up to the recent year? Previous studies have reported accelerated warming in the tropical Indian Ocean post the mid-1970s (eg: Kumar et al., 2009; Trenary and Han, 2008 Saji and Yamagata, 2003). Hence we have chosen the 1980s decade as the first decade for our analysis.

What basis you said that 2012 is the strongest IOD year during the study period. I feel

2006 is the strongest. Cross check it with DMI index. The reviewer is correct, based on the DMI the 2006 IOD is stronger than the IOD of 2012, we have modified accordingly in the revised manuscript.

P7, line lines 1-2 : Sentence is not correct. We have removed that sentence.

P8, line 20: Can you show time series of D20 in your boxes ? Unfortunately the sub-surface temperature data used in this analysis extends only 2012 and hence making use of this data is not feasible when the analysis is extended till 2018. So in the revised manuscript we are not making use of this data.

P8, line 10: conductive? We have replaced that word with the word "favorable" in the modified manuscript.

I think wrongly written Conclusion In your conclusion you have mentioned that asymmetrical warming favoring the formation of frequent positive IOD years. Is it your result ? If so, I have not seen this part discussed elsewhere in the ms. Which is very important.

We meant asymmetrical warming in the equatorial Indian Ocean region. The western equatorial Indian ocean region is warming at a rate faster than the eastern equatorial Indian Ocean. The increased warming of the colder western equatorial Indian Ocean (when compared to the eastern equatorial Indian Ocean) may result in the increased frequency of formation of IOD years. This has been reported previously by Roxy et al. (2014) in their study.

Why EAS shows higher warming than the western AS ? Can you show an SST trend of western AS in the same period? We have analyzed the trend of the SSTs of western Arabian Sea (WAS - 55E-65E, 8N-18N) and eastern Arabian Sea (EAS – 65E-7E, 8N-18N). It was found that the mean SSTs of EAS (28. 56 C) is higher than that of WAS (27.49 C) . The annual average SSTs showed an increasing trend of 0.57 C in the EAS whereas it was 0.46 C in the WAS, during the period 1981 to 2018. This means that

the zonal gradient of SSTs has increased in the Arabian Sea during the study period (See Fig. 3 & Fig. 4). Modelling studies by (eg: Decastro et al, 2016; Praveen et al., 2016) have shown that in a global warming scenario, the upwelling along the western Arabian becomes more intense. The increased upwelling off the coast of Africa result in reduced SST trends along the western Arabian Sea. (See fig. 4).

References:

Behrenfeld, M. J., O'Malley, R. T., Siegel, D. A., McClain, C. R., Sarmiento, J. L., Feldman, G. C., Milligan, A. J., Falkowski, P. G., Letelier, R. M. and Boss, E. S.: Climatedriven trends in contemporary ocean productivity, Nature., 444(7120), pp. 752-756, 2006.

Decastro, M., Sousa, M.C., Santos, F., Dias, J.M. and Gómez-Gesteira, M., 2016. How will Somali coastal upwelling evolve under future warming scenarios?. Scientific reports, 6, p.30137.

Kumar, M.R., Krishnan, R., Sankar, S., Unnikrishnan, A.S. and Pai, D.S., 2009. Increasing trend of "break-monsoon" conditions over India—role of ocean–atmosphere processes in the Indian Ocean.IEEE Geoscience and Remote Sensing Letters, 6(2), pp.332-336.

Parvathi, V., Suresh, I., Lengaigne, M., Ethé, C., Vialard, J., Levy, M., Neetu, S., Aumont, O., Resplandy, L., Naik, H. and Naqvi, S.W.A., 2017. Positive Indian Ocean Dipole events prevent anoxia off the west coast of India. Praveen, V., Ajayamohan, R.S., Valsala, V. and Sandeep, S., 2016. Intensification of upwelling along Oman coast in a warming scenario. Geophysical Research Letters, 43(14), pp.7581-7589.

Roxy, M.K., Ritika, K., Terray, P. and Masson, S., 2014. The curious case of Indian Ocean warming. Journal of Climate, 27(22), pp.8501-8509.

Saji, N.H. and Yamagata, T., 2003. Structure of SST and surface wind variability during Indian Ocean dipole mode events: COADS observations. Journal of Climate, 16(16),

pp.2735-2751.

Trenary, L.L. and Han, W., 2008. Causes of decadal subsurface cooling in the tropical Indian Ocean during 1961–2000. Geophysical Research Letters, 35(17).

Please also note the supplement to this comment:
https://www.biogeosciences-discuss.net/bg-2019-169/bg-2019-169-AC1-supplement.pdf

[Figure]

[Figure]

**Fig. 1.**

[Figure]

(NIOD−PIOD)          (PIOD−NEUTRAL)

**Fig. 2.**

[Figure]

[Figure]

**Fig. 3.**

[Figure]

**Fig. 4.**

---

## Author Comment (AC2) · 9 Aug 2019

We would like to thank the anonymous reviewer for the valuable comments and suggestions. The reply to reviewer's queries are listed below.

My major concern is that I do not understand what the motivation and purpose of the present study is. Neither do I understand what the authors want to convey to the reader. In the 11 lines of abstract there is no substantial result that is worth reporting. As indicated under general comment the first result is inconsequential as the decreasing trend in surface chlorophyll is not statistically significant. The 2nd result is already known as previous authors have already reported declining trend in the chlorophyll

concentration using long term data as well as model simulation

We have tried to address the chlorophyll trend for the period 1998 to 2018 during the Sep-Nov months when IOD is in its mature phase. We found that the chlorophyll trend in the CEAS box shows a decreasing trend that is statistically significant at the 95% confidence level. In the SEAS box, however there no statistically significant trend (Fig. 1). Close to the coast in the SEAS there are regions with positive chlorophyll trends whereas away from the coast chlorophyll mostly shows a decreasing trend. The increased warming trend in the eastern Arabian Sea is a major factor to influence the negative chlorophyll trend in the area CEAS. The chlorophyll concentration close to the coast in the SEAS is greatly influenced by local as well as remotely forced winds. Further studies are needed to ascertain the reasons for the increasing trend of chlorophyll concentration just off the south west coast of India and along the tip of the subcontinent. We have modified the text accordingly in the revised manuscript. We have also modified the abstract appropriately. We have also addressed the asymmetrical warming in the equatorial Indian Ocean region. The western equatorial Indian ocean region is warming at a rate faster than the eastern equatorial Indian Ocean. The increased warming of the colder western equatorial Indian Ocean (when compared to the eastern equatorial Indian Ocean) may result in the increased frequency of formation of IOD years. This has been reported by Roxy et al. (2014) in their study.

Why the authors have chosen only 3 tiny boxes close to the coast along the eastern boundary to examine the role of IOD.

We have increased the size of the boxes to extend further off coast and also reduced the number of boxes to 2 in the revised manuscript. The south eastern Arabian Sea is a major upwelling zone and a major source of several commercially important pelagic fish. Any long term changes to the chl-a concentration can adversely affect the pelagic fishery of the region.

October is a transition month from summer monsoon to winter monsoon. What is the

rationale in selecting only the month of October for the study? Why not consider the coastal upwelling months of June to September?

We had selected October as it is the month in which the strength of the IOD is maximum. The months Jun-Aug can be considered as the developing phase of the IOD whereas Sep-Nov can be considered as the mature phase of IOD. In the revised manuscript we have used the average of the months Sep-Nov, when IOD is in its mature phase, instead of October for analysis.

What is the role of EICC in bringing about the variability in chlorophyll concentra- tions? During fall (Oct-Dec) EICC flows equator-ward and moves around Sri Lanka and brings low saline nutrient rich water into the southeastern Arabian Sea.

How IOD, surface chlorophyll and Kelvin wave is connected? It is not clear from the manuscript. During a positive IOD anomalous easterlies are seen in the equatorial Indian Ocean. These easterlies forces upwelling Kelvin waves that propagate eastwards. Once these Kelvin waves reach the eastern equatorial Indian Ocean, they bifurcate into two. One part moves northward as a coastally trapped Kelvin wave and propagate along the Bay of Bengal boundary and finally reaches the eastern Arabian Sea thereby influencing the thermocline depth of the region. We have added this in the text of the revised manuscript.

Why study period was restricted to 2014 from 1998, when the chlorophyll data is available until 2018? We have now extended our study period from 2014 to 2018 in the revised manuscript.

In the study area (2 tiny boxes along the eastern boundary) data on D20 is mostly not available as could be seen from Figs. 8 & 9. In such case how reliable are the inferences that the authors draw based on these diagrams?

The D20 data that we used in this analysis extends only till 2012 and hence in the revised manuscript with extended period, we have not made of this data.

[Figure]

How the asymmetric warming favours formation of more frequent formation of positive IOD? The western equatorial Indian ocean (WEIO) is warming at a rate higher than that of eastern equatorial Indian Ocean (EEIO) and this results in the modification of the zonal gradient of temperature. The increased warming trend of the WEIO when compared to that of EEIO favors the development of frequent positive IOD years. This has been reported previously by Roxy et al. (2014) in their study.

What is/are the reason/s for the higher warming of the eastern Arabian Sea com- pared to the western Arabian Sea? This is not explained anywhere in the ms. Modelling studies by (eg: Decastro et al, 2016; Praveen et al., 2016) have shown that in a global warming scenario, the upwelling along the western Arabian Sea becomes more intense. The increased upwelling off the coast of Africa results in reduced SST trends along the western Arabian Sea. We have now included these points in the revised manuscript.

If the steady declining trend in surface chlorophyll in all the 3 small boxes along the eastern Arabian Sea is statistically insignificant

Now we have replaced the 3 small boxes with 2 larger boxes. In one box (CEAS) the negative trend of surface chla is statistically significant whereas in the other box (SEAS) the chla has marginally increased, though not statistically significant.

What is the purpose of lines 13 to 32 in page 3? It can be deleted as it do not contribute to the theme of the ms. We have deleted those lines as suggested by the reviewer.

What is "indigenous" IOD (line 19, page 2) We meant that IOD develops within the Indian Ocean basin.

What is the "marine biological activity"? (line 5, page 4) It is meant to be "marine biological productivity". We have corrected it in the manuscript.

In equations 3, 4 & 5, the notations for curl, zonal and meridional component of geostrophic current respectively are incorrect. We have corrected the notations in the

revised manuscript.

What is "standardized anomalies" Standardized anomalies are calculated by dividing anomalies by the climatological standard deviation.

Page 6: The authors cannot brush aside this by simply saying that our nterest is only in positive IOD. The authors need to address "Why Seas chlorophyll anomaly alone showed a negative value while others were positive" We have now modified the boxes and accordingly the explanations.

Lines 23-25, Page 6: What is the basis for this statement? If it is so why such drivers are not being considered during positive phase of IOD? We have removed this statement from the revised manuscript.

Line 30, Page 7: How this is possible? During transition the winds over the Arabian Sea is weak and variable We have rewritten this sentence in the manuscript.

Line 4, Page 8: What is "vertical component"? This is not Ekamn mass transport. Authors may like to consult standard Oceanographic text book. We have modified the sentence in the revised manuscript.

Lines 22-23, page 8: Upwelling is not the only process that brings about the up- sloping and down-sloping of isotherms. Modify. We have modified the manuscript.

Lines 5-10, Page 9: I cannot understand what authors wish to communicate? This is well documented by previous authors. How this is relevant in the present context. We have removed these sentences from the manuscript.

Lines 12-14, page 9: This is ambiguous. Authors need to provide a robust mech- anism that unambiguously addresses how WICC would impact change in chlorophyll.

The coastal Kelvin waves generated in the east coast of India drive a poleward coastal current (WICC) along the west coast of India. This current advects low-salinity water to the eastern Arabian Sea and inhibits convective mixing. This inhibition of mixing

reduces the entrainment of nutrients into the mixed layer and leads to low chlorophyll concentration. Previous studies (eg: Vijith et al, 2016) have shown how remotely forced coastal Kelvin waves impact the biology in the Arabian Sea.

References: Roxy, M.K., Ritika, K., Terray, P. and Masson, S., 2014. The curious case of Indian Ocean warming. Journal of Climate, 27(22), pp.8501-8509.

Vijith, V., Vinayachandran, P.N., Thushara, V., Amol, P., Shankar, D. and Anil, A.C., 2016. Consequences of inhibition of mixed‐layer deepening by the West India Coastal Current for winter phytoplankton bloom in the northeastern Arabian Sea. Journal of Geophysical Research: Oceans, 121(9), pp.6583-6603.

Please also note the supplement to this comment:
https://www.biogeosciences-discuss.net/bg-2019-169/bg-2019-169-AC2-supplement.pdf

——————————————————

[Figure]

[Figure]

[Figure]

Fig. 1.

---

## Author Comment (AC3) · 12 Aug 2019

**Results (modifications)**

It is well known that the upwelling along the west coast of India is influenced by local winds as well as remotely forcing (Yu et al., 1991; McCreary et al., 1993; Shankar and Shetye, 1997; Shankar et al., 2002). A modelling study by Suresh et al. (2016) has shown that winds near Sri Lanka drive 60% of seasonal sea level of Indian west coast where as the contribution from Bay of Bengal wind forcing is only 20%. They also pointed out that sea level signals forced by the winds near Sri Lanka extend westward into the eastern Arabian Sea with more than 50% contribution in the Lakshadweep high/low region. Negative seasonal sea level anomaly and associated thermocline shoaling in the southeastern Arabian Sea (Lakshdweep low region) during the summer monsoon brings nutrients near the surface causes phytoplankton bloom, and thus influences the food chain with a direct impact on the local fisheries (Madhupratap et al., 2001). A recent study by Suresh et al. (2018) showed that during positive IOD events downwelling Kelvin waves induce a positive sea level anomaly and a deep thermocline along the west coast of India very quickly (within days) during fall. Also, the equatorial easterlies force upwelling Kelvin waves that travel through the Bay of Bengal coastal waveguide to the west coast of India very slowly finally resulting in negative sea level anomaly in winter. The sea level anomaly along the west coast thus shifts from positive in fall to negative in winter during positive IOD events. Our results have shown that chlorophyll concentration is low along the south west coast of India during positive IOD years when compared to neutral and negative IOD years (Fig. 1) due to the presence of these downwelling Kelvin waves during Sep-Nov when IOD strength is at its peak. The maximum difference in chlorophyll concentration during a positive IOD year as against neutral/negative IOD year is seen from the tip of the subcontinent to below 12°N. This is in accordance with the findings of Suresh et al., (2018) who have shown that the equatorial easterly wind-stress anomalies during a positive IOD extend off the equator to approximately 10°N.

[Figure]

Fig .1 The difference in chlorophyll concentration in mg m$^{-3}$ for the Sep-Nov months during (left) NIOD and PIOD years and (right) PIOD and neutral IOD years

Our analysis has shown that the trend of the chlorophyll concentration in the CEAS is negative and statistically significant ($p < 0.05$) (Fig. 2 and Fig. 3). This is can be attributed to the increase in the SSTs resulting in enhanced stratification and reduction in nutrient exchange through vertical mixing (Behrenfeld et al., 2006, Capotondi et al., 2012). The chlorophyll trends may also be skewed due to the negative IOD of 2016, which is the strongest negative IOD recorded since 1980 (Lu et al., 2018). The SSTs have increased by 0.57 °C (0.46 °C) during the 38 year period 1981 to 2018 in the eastern (western) Arabian Sea area (Fig. 4 and Fig. 5). In the SEAS, the regions away from the coast shows negative chlorophyll trends whereas regions very close to the close to the coast show positive chlorophyll trends. But there is no significant trend in the chlorophyll concentration in SEAS area as a whole ($p = 0.996$). It is also interesting to note that within the SEAS area, SST warming trend is lower close to the coast where the chlorophyll trend is positive. Prakash et al. (2012) had shown that there is no significant trend in the chlorophyll in a small area off the south west coast of India covered by the area SEAS of our analysis. The increase in surface chlorophyll concentration very close to the coast is due to the increased wind stress over this region. In addition to the reduced positive trend of SSTs close to the south west coast of India (east of 70°E (Fig. 4), several modelling studies have shown that the withdrawal of the Indian summer monsoon season is getting delayed in a warming environment (eg: Jayasankar et al., 2015), resulting in increased upwelling favorable conditions during Sep-

Nov months.

[Figure]

Fig. 2 Chlorophyll trend in mg m$^{-3}$ for the Sep-Nov months durring the period 1981-2018. area 1 (65° E-75° E 12° N-18° N) represents the central eastern Arabian Sea (CEAS), area 2 (65° E-76° E, 7° N-12° N) represents the south eastern Arabian Sea (SEAS).

[Figure]

Fig. 3 a) The surface chlorophyll trend in mg m$^{-3}$ in the eastern Arabian Sea area (65E-75E, 8N-18N) and b) chlorophyll trend in mg m$^{-3}$ in the western Arabian Sea (55E-65E, 8N-18N) during the study period 1981-2018.

[Figure]

Fig. 4 The map of tropical Indian Ocean highlighting the study area. WAS represents the western Arabian Sea (55° E–75° E , 8° N–18° N, EAS represents the eastern Arabian Sea (65° E–75° E, 8°

N–18° N). The annual SST trend (°C) during the period 1981 to 2018 is also shown in the image.

[Figure]

Fig. 5 a) The SST trend in the eastern Arabian Sea area (65E-75E, 8N-18N) and b) SST trend in the western Arabian Sea (55E-65E, 8N-18N) during the study period 1981-2018.

**References:**

Behrenfeld, M. J., O'Malley, R. T., Siegel, D. A., McClain, C. R., Sarmiento, J. L., Feldman, G. C., Milligan, A. J., Falkowski, P. G., Letelier, R. M. and Boss, E. S.: Climate-driven trends in contemporary ocean productivity, Nature., 444(7120), pp. 752-756, 2006.

Capotondi, A., Alexander, M.A., Bond, N.A., Curchitser, E.N. and Scott, J.D., 2012. Enhanced upper ocean stratification with climate change in the CMIP3 models. *Journal of Geophysical Research: Oceans*, *117*(C4).

Jayasankar, C.B., Surendran, S. and Rajendran, K., 2015. Robust signals of future projections of Indian summer monsoon rainfall by IPCC AR5 climate models: Role of seasonal cycle and interannual variability. *Geophysical Research Letters*, *42*(9), pp.3513-3520.

Lu, B., Ren, H.L., Scaife, A.A., Wu, J., Dunstone, N., Smith, D., Wan, J., Eade, R., MacLachlan, C. and Gordon, M., 2018. An extreme negative Indian Ocean Dipole event in 2016: dynamics and predictability. *Climate dynamics*, *51*(1-2), pp.89-100.

Madhupratap, M., Gopalakrishnan, T.C., Haridas, P. and Nair, K.K.C., 2001. Mesozooplankton biomass, composition and distribution in the Arabian Sea during the fall intermonsoon: implications of oxygen gradients. *Deep Sea Research Part II: Topical Studies in Oceanography*, *48*(6-7), pp.1345-1368.

Prakash, P., Prakash, S., Rahaman, H., Ravichandran, M. and Nayak, S., 2012. Is the trend in chlorophyll-a in the Arabian Sea decreasing?. *Geophysical Research Letters*, *39*(23).

Suresh, I., Vialard, J., Izumo, T., Lengaigne, M., Han, W., McCreary, J. and Muraleedharan, P.M., 2016. Dominant role of winds near Sri Lanka in driving seasonal sea level variations along the west coast of India. *Geophysical Research Letters*, *43*(13), pp.7028-7035.

Suresh, I., Vialard, J., Lengaigne, M., Izumo, T., Parvathi, V. and Muraleedharan, P.M., 2018. Sea level interannual variability along the west coast of India. *Geophysical Research Letters*, *45*(22), pp.12-440.